# Efficacy and Adverse Events in Metastatic Melanoma Patients Treated with Combination BRAF Plus MEK Inhibitors Versus BRAF Inhibitors: A Systematic Review

**DOI:** 10.3390/cancers11121950

**Published:** 2019-12-05

**Authors:** Austin Greco, Danish Safi, Umang Swami, Tim Ginader, Mohammed Milhem, Yousef Zakharia

**Affiliations:** 1Department of Internal Medicine, University of Iowa, Iowa City, IA 52242, USA; danish-safi@uiowa.edu; 2Department of Internal Medicine, Division of Hematology and Oncology, University of Iowa, Iowa City, IA 52242, USA; umang.swami@hci.utah.edu (U.S.); mohammed-milhem@uiowa.edu (M.M.); yousef-zakharia@uiowa.edu (Y.Z.); 3Department of Internal Medicine, Division of Oncology, Huntsman Cancer Institute, University of Utah, Salt Lake City, UT 84112, USA; 4Holden Comprehensive Cancer Center, University of Iowa, Iowa City, IA 52242, USA; tim.ginader@gmail.com; 5Department of Biostatistics, College of Public Health, University of Iowa, Iowa City, IA 52242, USA

**Keywords:** adverse drug event, treatment outcome, melanoma, BRAF inhibition, MEK inhibition, encorafenib, binimetinib, dabrafenib, trametinib, cobimetinib, vemurafenib

## Abstract

We reviewed the literature to assess the efficacy and risk of constitutional, cardiac, gastrointestinal, and dermatological toxicities of combined BRAF plus MEK inhibitors versus BRAF inhibitors alone in patients with metastatic melanoma with BRAF mutations. Searches were conducted in PubMed, Cochrane Database of Systematic Reviews, Google scholar, ASCO, Scopus, and EMBASE for reports published from January 2010 through March 2019. Efficacy, including progression-free survival (PFS) and overall survival (OS) rates, were assessed by hazard ratio (HR); objective response rates (ORR) were assessed by odds ratio (OR). The randomized clinical trials (RCTs) with comparison to vemurafenib monotherapy were included to determine constitutional, gastrointestinal, cardiac, and dermatological toxicities using PRISMA statistical analysis with relative risk (RR) for equal comparison to avoid inclusion bias. Five RTCs comprising 2307 patients were included to assess efficacy, while three of the five RCTs comprising 1776 patients were included to assess adverse events. BRAF plus MEK inhibitor combination therapy demonstrated overall better efficacy compared to BRAF inhibitor monotherapy. Combination therapies appear to have favorable dermatologic side effect profiles, similar constitutional and cardiac profiles, and slightly worse gastrointestinal profiles compares to monotherapy regimens.

## 1. Introduction

Melanoma is the fifth most common cancer in both men and women in the United States (US). Survival rates depend largely on the stage of the disease at diagnosis, with 5-year survival rate ranging from approximately 98% for localized disease to only about 23% for metastatic disease with previous traditional chemotherapy regimens [1]. Until 2011, treatment for metastatic disease most commonly included chemotherapy with alkylating agents such as dacarbazine and temozolomide, and/or immune-stimulatory cytokines such as interleukin-2 which provided a small overall survival benefit [2].

Melanomas have a range of gene and protein alterations which are susceptible to therapies targeting the mitogen-activated protein kinase (MAPK) pathway. About 65% of melanomas contain mutations in the RAS–RAF–MEK–ERK pathway. Furthermore, approximately 40% harbor the BRAF V600 activating mutation [3,4]. Pharmacologic inhibition of the MAPK pathway has proven to be a major advancement in the treatment of metastatic melanoma.

BRAF inhibitors, dabrafenib and vemurafenib, as monotherapy have demonstrated significant improvement in overall survival (OS), progression-free survival (PFS), and objective response rates (ORR) compared to traditional chemotherapy [5,6,7]. However, early development of resistance due to multiple mechanisms, including BRAF amplification or downstream activating mutations in the MAPK pathway, is a matter of concern [8]. Addition of MEK inhibitors such as trametinib and cobimetinib to BRAF inhibitors has shown to improve survival rates in multiple phase II and III randomized controlled trials, and has now became the standard of care [9,10,11,12]. Recently, in phase III COLUMBUS trial encorafenib and binimetinib, a new BRAF–MEK inhibitor combination has demonstrated survival benefit as compared to single agent BRAF inhibitors [13].

Apart from survival benefit, another area of current focus is toxicity profile. Treatment decisions are not only guided by survival data, but also depend on drug toxicities, which can impair quality of life. To our knowledge, only one meta-analysis has compared survival data and adverse events between combination BRAF and MEK inhibition therapy versus BRAF inhibition monotherapy. Herein, we evaluate constitutional, gastrointestinal, cardiac, and dermatological adverse events (AEs) in BRAF and MEK inhibition combination therapy versus BRAF inhibition monotherapy.

## 2. Methods

### 2.1. Literature Search

This systematic review was conducted in accordance with the PRISMA (Preferred Reporting Items for Systematic Reviews and Meta-Analyses) guidelines [14]. Two separate physicians performed independent reviews of the literature in the English language to include publications from 1 January 2010 through 7 March 2019. Electronic searches were performed using PubMed, Cochrane Database of Systematic Reviews, ASCO, and EMBASE. The literature search strategy was designed to identify the most recent reports of phase I, II, or III clinical trials with direct comparison of survival data and adverse events between MEK and BRAF inhibition combination therapy versus BRAF inhibition monotherapy in unresectable stage IIIB, IIIC, or metastatic cutaneous melanoma with BRAF V600E or BRAF V600K mutation. Search terms with Boolean operators (AND, OR) were as follows: “melanoma,” OR “melanomas”; AND; “vemurafenib,” OR “dabrafenib,” OR “encorafenib,” OR “braf BRAF inhibition,” OR “braf inhibitor” OR “braf inhibitors”; AND; “mek inhibition” OR “mek inhibition,” OR “mek inhibitors” OR “cobimetinib,” OR “GDC-0973” OR “trametinib,” OR “binimetinib,” OR “MEK162”; with a filter for clinical trial. Each phrase or term was included in the search algorithm as medical subject headings or to be identified in the title or abstract. Search results were then screened for survival and AE data, and treatment adherence to US Food and Drug Administration dosing guidelines.

### 2.2. Literature Selection Criteria

All identified articles using the aforementioned parameters were then screened using inclusion criteria: (1) Prospective phase I, II, and III clinical trials in the English language; (2) clinical investigation in patients with unresectable stage IIIB, IIIC, or metastatic cutaneous melanoma with BRAF V600E or BRAF V600K mutation [10]; (3) direct comparison between MEK and BRAF inhibition combination therapy versus BRAF inhibition monotherapy study groups; (4) survival data available; (5) adverse events data available; and (6) therapy starting dose was at US FDA guidelines. Exclusion criteria were also used to screen for appropriate studies: (1) Study groups receiving ongoing therapy other than MEK and/or BRAF inhibition and (2) publications with more up-to-date data from same National Clinical Trial (NCT).

### 2.3. Data Extraction

Two physician investigators independently extracted and compared data. Any discrepancies were re-reviewed and a consensus was reached between the two investigators. Primary outcomes included: OS, PFS, and ORR. Secondary outcomes included AE rates with emphasis on constitutional events (pyrexia, fatigue, headache, myalgia, cough, decreased appetite), cardiac (peripheral edema, hypertension), gastrointestinal (constipation, nausea, vomiting, diarrhea, increased aspartate aminotransferase (AST), and increased alanine aminotransferase (ALT)), and dermatological (rash, hand-foot syndrome, hyperkeratosis, skin papilloma, alopecia, photosensitivity). Adverse events in these categories were excluded if data for these were omitted in one or more of the studies used in the analysis. AEs were categorized as any grade, low grade (characterized as grades 1–2), or high grade (characterized as grades 3–4). Grade 5, or death, was not collected for adverse events due to extremely low incidences in categories of adverse events being studied. In the study performed by Flaherty, only data from part C and the study groups receiving full U.S. FDA approved starting doses of therapies were included, as this was the only section meeting inclusion criteria.

### 2.4. Definition of Outcomes

Primary outcomes were defined according to RECIST version 1.1. PFS was defined as time from date of randomization to the date of first documented disease progression or death from any cause. ORR was defined as complete or partial response. Other primary outcomes include OS, defined as median duration of survival in months from date or randomization, and OS at 1 year defined as percentage of survival 1 year following randomization.

Definition and severity of adverse events were graded using the National Cancer Institute’s Common Terminology Criteria for Adverse Events, version 4.0.

### 2.5. Statistical Analysis

All statistical analyses were performed in R with the “rmeta” and “metaphor” packages. The difference in adverse events between combination therapy and monotherapy were estimated using relative risk (RR; increased risk of adverse event in combination therapy vs. monotherapy groups). Adverse events were summarized using a DerSemonion–Laird random effects model due to the large heterogeneity between studies, as evidenced by significant Cochran-Q tests and large I2 for the majority of the adverse event models. All adverse events were analyzed for all data available. Subgroup analyses for grade 3–4 adverse events were performed if incidence was high enough for appropriate calculation.

To compare adverse event data between groups more directly and to avoid inclusion and/or selection bias, only studies comparing combination therapy to vemurafenib monotherapy were analyzed.

PFS and OS were estimated using hazard ratios (HR; increased risk of death in combination therapy vs. monotherapy groups). ORR was measured using odds ratio (OR). Survival outcomes and ORR were summarized using a fixed effect model due to low heterogeneity.

### 2.6. Publication Bias

Publication biased was assessed using funnel plots displayed in Figure 1A–C. With only five studies, however, these plots are lacking adequate power in testing asymmetry.

## 3. Results

### 3.1. Study Selection

We identified 1382 potential articles after initial searches of all databases. After removal of duplicates, inclusion and exclusion criteria were used to analyze titles, abstracts, and full text, resulting in 14 papers. Lastly, reports of the same randomized control trial by NCT number were sorted and only the most recent was included. Five randomized clinical trial (RCT) studies remained for analysis, which are outlined in Figure 2. Of the resulted, there were four phase III studies and one phase I/II study [9,10,11,12,13].

The five studies comprised 2307 participants that were analyzed. For the study by Flaherty et al., only data within inclusion and exclusion criteria were used. This included two study groups of part C to only include US FDA guideline starting dosing of trametinib [10]. Therefore, only 108 of 162 patients from part C of this study were included in the analysis. For the study by Dummer, the participants in the monotherapy group treated with encorafenib were excluded [13]. The monotherapy group treated with vemurafenib was used in the analysis to compare to the combination group, as this monotherapy was commonly used in other phase III trials included and could offer superior direct comparison of combination therapy between studies. The remainder of the studies had all groups included in the meta-analysis. Characteristics of these trials are shown in Table 1.

### 3.2. Survival Outcomes

Each of the five studies analyzed revealed improved outcomes in patients receiving combination therapy vs. monotherapy. As demonstrated in Figure 3A, overall survival was improved in groups receiving combination therapy. When compared to vemurafenib monotherapy, as performed in the studies by Dummer, Robert, and Ascierto, encorafenib plus binimetinib had a hazard ratio (HR) of 0.76 (95% Confidence Internal (CI) 0.58–0.98), dabrafenib plus trametinib had a HR of 0.69 (95% CI 0.53–0.89), and vemurafenib plus cobimetinib had a HR of 0.70 (95% CI 0.55–0.90), all showing significant improvement in overall survival. When compared to dabrafenib monotherapy, as performed in the study by Long, dabrafenib plus trametinib had a HR of 0.71 (95% CI 0.55–0.92), also indicating a survival benefit in the combination group. In the trial by Flaherty, overall survival (OS) was not reached by the time on analysis. Hazard ratio for OS for all studies combined was 0.71 (95% CI 0.63–0.81) and Cochran’s Q for OS was 0.31 (*p* = 0.96), revealing overall improved overall survival in patients receiving combination therapy compared to monotherapy.

PFS was also improved in groups receiving combination therapy compared to groups receiving monotherapy, as demonstrated in Figure 3B. When compared to vemurafenib monotherapy, as performed in the studies by Dummer, Robert, and Ascierto, encorafenib plus binimetinib had a hazard ratio (HR) of 0.54 (95% CI 0.41–0.71), dabrafenib plus trametinib had a HR of 0.56 (95% CI 0.46–0.69), and vemurafenib plus cobimetinib had a HR of 0.59 (95% CI 0.47–0.73), respectively. When compared to dabrafenib, as performed in the study by Long and Flaherty, dabrafenib plus trametinib had a HR of 0.67 (95% CI 0.53–0.84) and 0.39 (95% CI 0.25–0.62) for the individual studies, respectively. Hazard ratio for PFS for all studies combined was 0.58 (95% CI 0.52–0.64) and Cochran’s Q was 4.82 (*p* = 0.31).

Similarly, as noted in Figure 3C, objective response rate (ORR) was superior in each group receiving combination therapy compared to monotherapy. In all studies combined, there was an odds ratio (OR) of 2.02 (95% CI 1.70–2.42), indicating significant improvement in ORR for patients receiving combination therapy. There did not seem to be any significant difference in efficacy data between groups receiving combination therapy.

### 3.3. Pyrexia and Other Constitutional Adverse Events

To compare adverse event data between groups treated with combination therapy more directly, only studies comparing combination therapy to vemurafenib monotherapy were analyzed [9,12,13]. These included the studies by Dummer, Robert, and Ascierto, which evaluated the combination groups of encorafenib plus binimetinib, dabrafenib plus trametinib, and cobimetinib plus vemurafenib, respectively.

Constitutional signs and symptoms are some of the most frequent adverse events for BRAF and MEK inhibition combination therapy and BRAF inhibition monotherapy. Pyrexia was the most prevalent in the study by Robert et al., with an incidence of 53% (184/350) in the group treated with dabrafenib plus trametinib and 21% (73/349) in the group treated with vemurafenib alone [12]. 

For pyrexia, analysis of the three studies comparing combination therapy to vemurafenib monotherapy revealed a combined relative risk (RR) of 1.27 (95% CI 0.59–2.73), showing no significant difference between groups (Figure 4). However, encorafenib plus binimetinib in the Dummer study had an RR of 0.65 (95% CI 0.45–0.95), revealing a significantly smaller risk of pyrexia as compare to monotherapy. Inversely, dabrafenib plus trametinib in the Robert study had an RR of 2.51 (95% CI 2.00–3.15), revealing a significantly higher risk of pyrexia in the combination group. Vemurafenib and cobimetinib in the Ascierto study did not show any significant difference in risk of pyrexia between groups. It can also be inferred that encorafenib plus binimetinib had a significantly smaller risk of pyrexia than the combination of dabrafenib plus trametinib.

There was no difference overall for fatigue, with a combined relative risk (RR) of 0.97 (95% CI 0.83–1.13), as shown in Appendix A. For headache, as shown in Appendix A, a combined relative risk (RR) of 1.21 (95% CI 1.01–1.48) showed a significantly higher risk of developing this adverse event the combination group. Individually, dabrafenib plus trametinib had an RR of 1.31 (95% CI 1.01–1.69), revealing a significantly higher risk of headache in the combination group compared to vemurafenib, while the other two combination groups showed no significant difference. 

There was no difference overall for cough, with a combined relative risk (RR) of 1.24 (95% CI 0.64–2.39), as shown in Figure 5. However, dabrafenib plus trametinib had an RR of 2.02 (95% CI 1.38–2.97), revealing a significantly larger risk of cough as compare to monotherapy. While the other two combination groups had no significant difference compared to monotherapy. Importantly, vemurafenib plus cobimetinib had a significantly smaller risk of cough than the combination of dabrafenib plus trametinib, as confidence intervals for respective relative risks are not overlapping and are compared to a monotherapy-like group.

Pertaining to decreased appetite in Appendix A, there was no overall difference between combination therapy and monotherapy, with a relative risk (RR) of 0.66 (95% CI 0.41–1.05). However, encorafenib plus binimetinib had an RR of 0.43 (95% CI 0.25–0.75) compared to the vemurafenib group, revealing a significantly smaller risk of decreased appetite as compare to monotherapy. A similar finding was found in the dabrafenib plus trametinib, with an RR of 0.60 (95% CI 0.42–0.85).

For adverse events of fatigue in Figure 6 and myalgia in Appendix A, no significant difference could be demonstrated overall or in individual studies between combination therapy groups and monotherapy groups. Combined relative risk (RR) of 0.97 (95% CI 0.83–1.13) and 1.24 (95% CI 0.64–2.39) were found for fatigue and myalgia, respectively.

### 3.4. Gastrointestinal Adverse Events

Nausea and diarrhea are the most common gastrointestinal AEs. Ascierto et al. reported nausea incidence rates of 43% (105/247 patients) on cobimetinib plus vemurafenib therapy and 26% (64/248) in the vemurafenib monotherapy group [9]. The incidence in the combination group of dabrafenib plus trametinib was found to be only 35% (121/350) in the trial by Robert [12]. In all of the trials analyzed, Ascierto was the only trial to show a statistically significant greater occurrence of nausea in the combination group of cobimetinib plus vemurafenib as compared to the vemurafenib monotherapy group. It can also be said that cobimetinib plus vemurafenib (RR: 1.65, 95% CI 1.28–2.13) had a significantly higher risk of nausea than the combination of dabrafenib plus trametinib (RR: 0.97, 95% CI 0.79–1.18), as confidence intervals for respective relative risks are not overlapping and are compared to a monotherapy-like group, as shown in Appendix A. Similarly, for diarrhea, the combination of cobimetinib plus vemurafenib had an incidence of 61% (150/247) [9]. When comparing to the other combination groups in Figure 7, cobimetinib plus vemurafenib therapy (RR: 1.84, 95% CI 1.50–2.25) was shown to have an increased risk of diarrhea compared to encorafenib plus binimetinib (RR: 1.08, 95% CI 0.82–1.42) therapy or dabrafenib plus trametinib therapy (RR: 0.85, 95% CI 0.70–1.05), as confidence intervals for respective relative risks are not overlapping and are compared to a monotherapy-like group.

For decreased appetite, there was no significant difference overall when comparing combination therapy to monotherapy, with an RR of 0.66 (95% CI 0.41–1.05) (Appendix A). Similarly, for constipation, there was an overall relative risk of 1.83 (95% CI 0.9–3.62), showing no significant difference (Appendix A). As shown in Appendix A, nausea had a relative risk of 1.24 (95% CI 0.90–1.69), showing no significant difference overall between combination therapy and monotherapy. However, cobimetinib plus vemurafenib combination alone did have a lower incidence of nausea compare to vemurafenib, with a relative risk of 1.65 (95% CI 1.28–2.13). As shown in Appendix A, there was a much higher incidence in the combination group compare to monotherapy, with an overall RR of 1.91 (95% CI 1.56–2.33).

For ALT and AST elevation (Appendix A), there was no overall significant difference when comparing combination therapy groups. However, cobimetinib plus vemurafenib did show a significantly higher risk as compared to vemurafenib monotherapy, with a relative risk of 1.48 (CI 95% 1.06–2.08) and 1.94 (CI 95% 1.31–2.89) for ALT and AST elevation, respectively.

### 3.5. Cardiac Adverse Events

For cardiac adverse events, hypertension had the highest occurrence. For example, in the study by Robert, the combination group of dabrafenib plus trametinib had an incidence of 26% for all grade hypertension (92/350) [12]. Overall, there was no significant difference in all studies combined for hypertension in combination therapy versus monotherapy, with an RR of 1.25 (95% CI 0.86–1.83), as shown in Appendix A. However, cobimetinib plus vemurafenib combination showed a significantly higher incidence when compared to vemurafenib monotherapy (RR: 1.96, 95% CI 1.18–3.26). There was no change noted between combination groups. For peripheral edema, there was no overall difference between combination groups and monotherapy groups (RR: 1.13, 95% CI 0.80–1.59) (Appendix A).

### 3.6. Dermatology Adverse Events

Rash is a common adverse event with these therapies. For example, encorafenib plus binimetinib combination therapy was found to have an incidence of 14% (27/192), dabrafenib plus trametinib 22% (76/350), and cobimetinib plus vemurafenib of 41% (101/247) [9,12,13]. Overall, as shown in Figure 8, there was no significant difference in the development of rash in combination therapy versus monotherapy, with a relative risk (RR) of 0.65 (95% CI 0.37–1.14). There was statistical significance in in the encorafenib plus binimetinib group and the dabrafenib plus trametinib combination therapies when compared to monotherapy, with a relative risk of 0.48 (95% CI 0.32–0.73) and 0.51 (95% CI 0.40–0.64), respectively. It can also be said that encorafenib plus binimetinib and dabrafenib plus trametinib combination therapies have a lower risk of developing rash compared to vemurafenib plus cobimetinib. For hyperkeratosis, all combination groups had a significantly decreased risk compared to monotherapy, with a relative risk of 0.26 (95% CI 0.09–0.76) (Appendix A). Furthermore, encorafenib plus binimetinib combination therapy had a higher risk compared to dabrafenib plus trametinib combination therapy. Similarly, for skin papilloma and alopecia, combination groups had a significantly decreased risk compared to monotherapy, with a relative risk of 0.26 (95% CI 0.09–0.76) and 0.31 (95% CI 0.15–0.66), respectively (Appendix A). Dabrafenib plus trametinib combination therapy was found to be superior, with a lower risk of skin papilloma and alopecia compared to vemurafenib plus cobimetinib and encorafenib plus binimetinib combination therapies. For hand–foot syndrome, there was found to be no significant difference overall between combination and monotherapy, with a relative risk of 0.52 (95% CI 0.13–2.00) (Appendix A). Notably, dabrafenib plus trametinib was superior to monotherapy (RR: 0.16, 95% CI 0.09–0.28), as well as the vemurafenib plus cobimetinib combination group. Lastly, for photosensitivity, while there was no overall significant difference between combination therapy and monotherapy (RR: 0.38, 95% CI 0.06–2.24), it can be said that encorafenib plus binimetinib and dabrafenib plus trametinib combination therapies have a lower risk of photosensitivity compared to vemurafenib plus cobimetinib (Appendix A).

## 4. Discussion

Approximately 65% of patients with metastatic melanoma have RAS–RAF–MEK–ERK pathway mutations, and 40% have BRAF V600 activating mutations [3,4]. Utilizing this pathway with targeted inhibitors is now a mainstay of therapy.

Currently, there are three BRAF–MEK inhibition combination therapies approved for first-line treatment of metastatic melanoma with BRAF V600 activating mutations, including: Vemurafenib plus cobimetinib, dabrafenib plus trametinib, and encorafenib plus binimetinib [15]. The use of combination therapy of BRAF and MEK inhibition has proven to be superior in terms of survival rates compared to BRAF inhibition alone [9,10,11,12,13]. Mechanistically, this is thought to be secondary to reduced resistance and decreased paradoxical activation of other proteins in the pathway [16,17,18,19,20]. However, there is concern for synergistic toxic effects with multiple drugs targeting the same signaling pathway, making adverse event profiles paramount in choosing the appropriate therapy for patients.

This review was performed on five studies with a total of 2307 participants, where direct comparison between combination therapy and monotherapy was performed. For survival data, all studies were analyzed. To more directly compare toxicities between groups treated with combination therapy, only combination groups compared to vemurafenib monotherapy were statistically analyzed. These included the combination therapies of encorafenib plus binimetinib, dabrafenib plus trametinib, and cobimetinib plus vemurafenib [9,12,13].

As previous studies have demonstrated, combination therapy shows statistically significant improvement in survival data, including overall survival (OS), progression free survival (PFS), and objective response rate (ORR) [21]. This continues to be evident in our systematic review. When combining data from the studies analyzed, there was found to be significant improvement in OS, PFS, and ORR in groups receiving combination therapy compared to monotherapy. Encorafenib plus binimetinib and dabrafenib plus trametinib showed statistically significant overall survival compared to vemurafenib [10,12,13]. However, confidence intervals of each group were overlapping, indicating no change in overall survival between the combination groups themselves. When compared to dabrafenib monotherapy, as performed in the study by Long, dabrafenib plus trametinib also showed superior overall survival when compared to dabrafenib monotherapy [11]. The same can be said for each trial comparing combination therapy to monotherapy for PFS and ORR [10,11,12,13]. And again, when comparing trials, confidence intervals of each group overlap, indicating no difference in OS, PFS, or ORR between combination therapy groups, can be identified. Further study is necessary to provide insight into the most appropriate combination therapy based on survival data. Our hope is that this study will aid in selecting therapy regimens with known comparable survival data and potentially differing toxicity profiles.

The etiology of pyrexia is not known for patients treated with BRAF inhibition and/or MEK inhibition therapies; however, this very common adverse event is often dose- and therapy-limiting [22]. Combined analysis of the three studies comparing combination therapy to vemurafenib monotherapy revealed no significant difference between combination therapy and monotherapy. However, encorafenib plus binimetinib showed a significantly smaller risk of pyrexia compared to monotherapy. Inversely, dabrafenib plus trametinib revealed a significantly higher risk of pyrexia in the combination group. Furthermore, when comparing studies, encorafenib plus binimetinib had a significantly lower incidence of pyrexia compared to dabrafenib plus trametinib. Combination dabrafenib plus trametinib also demonstrated a significantly increased risk of headache compared to monotherapy; however, no significant difference was found between combination groups.

For cough, while there was no difference overall between combination and monotherapy, dabrafenib plus trametinib had a significantly larger risk of cough as compared to monotherapy. Furthermore, vemurafenib plus cobimetinib had a significantly smaller risk of cough than the combination of dabrafenib plus trametinib. For adverse events of fatigue and myalgia, no significant difference could be demonstrated overall or in individual studies between combination therapy groups and monotherapy groups.

In terms of gastrointestinal (GI)-related events, MEK inhibitors have been previously associated with multiple GI-related adverse events, which can be easily reversed by temporary discontinuation and restarting at a lower dose [23]. Constipation was significantly more common in the encorafenib plus binimetinib and dabrafenib plus trametinib combination groups compared to vemurafenib monotherapy. Vemurafenib plus cobimetinib appears to have a significantly higher risk of nausea than monotherapy and dabrafenib plus trametinib combination therapy. Notably, all combination groups had higher incidences of vomiting compared to monotherapy.

Cardiac-related adverse events show that there is no difference in peripheral edema and hypertension overall in combination group in comparison to monotherapy groups. For hypertension, cobimetinib plus vemurafenib combination showed a significantly higher incidence when compared to vemurafenib monotherapy. The etiology of hypertension in these therapies is usually explained by two differing mechanisms. One mechanism is interference of renin–angiotensin regulation due to inhibition of BRAF and MEK pathway [24], and the second mechanism is due to decreased nitric oxide (NO) production due to inhibition of MAPK pathway [25].

Dermatologic reactions are well studied in BRAF and MEK inhibition therapies. Clinically, dermatologic adverse events have been shown to be more common in BRAF inhibitor monotherapy compared to BRAF and MEK inhibitor combination therapy [9,12,13]. BRAF inhibition monotherapy may lead to paradoxical activation of the MAPK pathway, which may explain why combination therapy demonstrates superior toxicity profiles for dermatologic adverse events [26]. Although overall there was no significant difference in incidence of rash between combination therapy and monotherapy, individual studies analyzed showed a lower incidence in encorafenib plus binimetinib and dabrafenib plus trametinib when compared to monotherapy. The same can be said for hand–foot syndrome and photosensitivity. For hyperkeratosis, skin papilloma, and alopecia, all combination therapies showed significantly lower incidence when compared to monotherapy.

## 5. Conclusions

In conclusion, combination BRAF inhibition and MEK inhibition therapy when compared to BRAF inhibition monotherapy demonstrated significant improvement in OS, PFS, and ORR rates in trials reported to date. Combination therapies of encorafenib plus binimetinib, dabrafenib plus trametinib, and cobimetinib plus vemurafenib show clear survival benefits when compared to vemurafenib or dabrafenib alone. Our analysis was unable to show any significant differences between combination groups.

Although survival outcomes are well known, this study allows for better comparison of adverse events between therapies. For constitutional adverse events, encorafenib plus binimetinib would be a superior choice compared to dabrafenib plus trametinib or monotherapy for pyrexia. Encorafenib plus binimetinib or dabrafenib plus trametinib are both superior to monotherapy in avoiding decreased appetite. For other gastrointestinal toxicities, monotherapy regimens seem to have favorable adverse event incidences. All combination groups had higher incidences of vomiting compared to monotherapy. For diarrhea, vemurafenib plus cobimetinib is the least favorable compared to the monotherapy and the other combination groups and should be avoided in patients with concerns of diarrhea. Likewise, vemurafenib plus cobimetinib may be the least favorable choice if there is concern of hypertension. Most dermatologic adverse events have a lower occurrence with combination therapy regimens compared to monotherapy. Most notably, encorafenib plus binimetinib or dabrafenib plus trametinib should be favored in patients with concern of rash. We feel that adverse event profiles discussed in this systematic review will offer guidance into choosing the appropriate therapy regimen for patients.

## Figures and Tables

**Figure 1 cancers-11-01950-f001:**
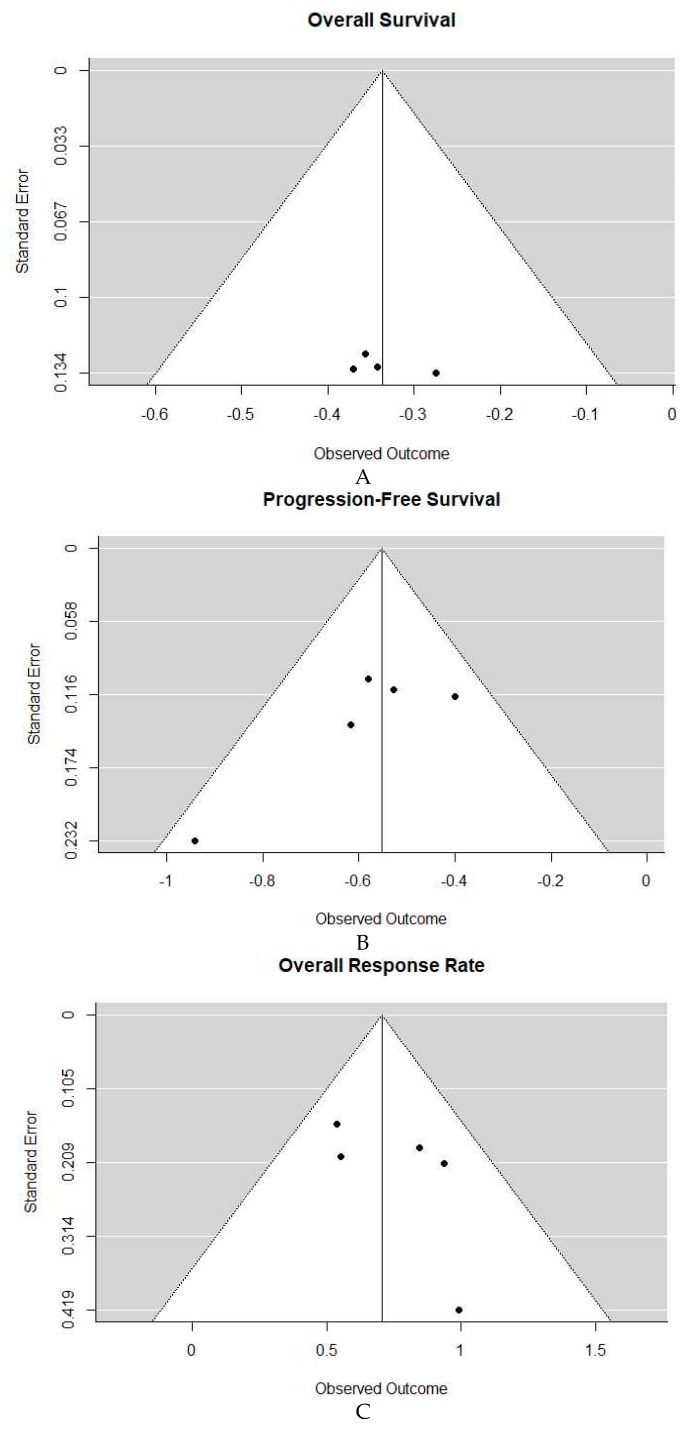
(**A**). Funnel plot overall survival (OS). (**B**) Funnel plot progression-free survival (PFS). (**C**) Funnel plot overall response rate (ORR).

**Figure 2 cancers-11-01950-f002:**
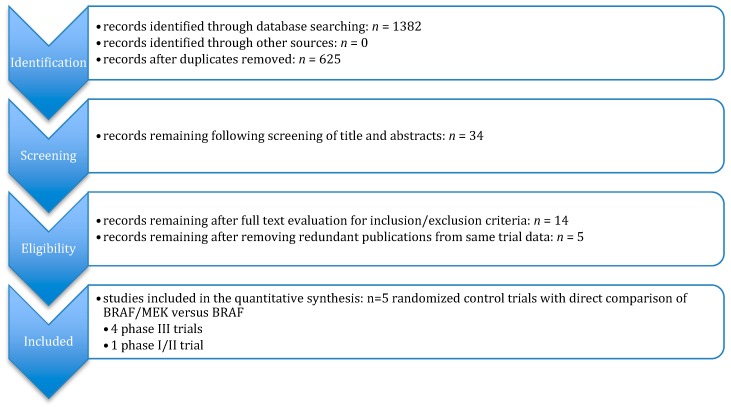
Study selection flow chart.

**Figure 3 cancers-11-01950-f003:**
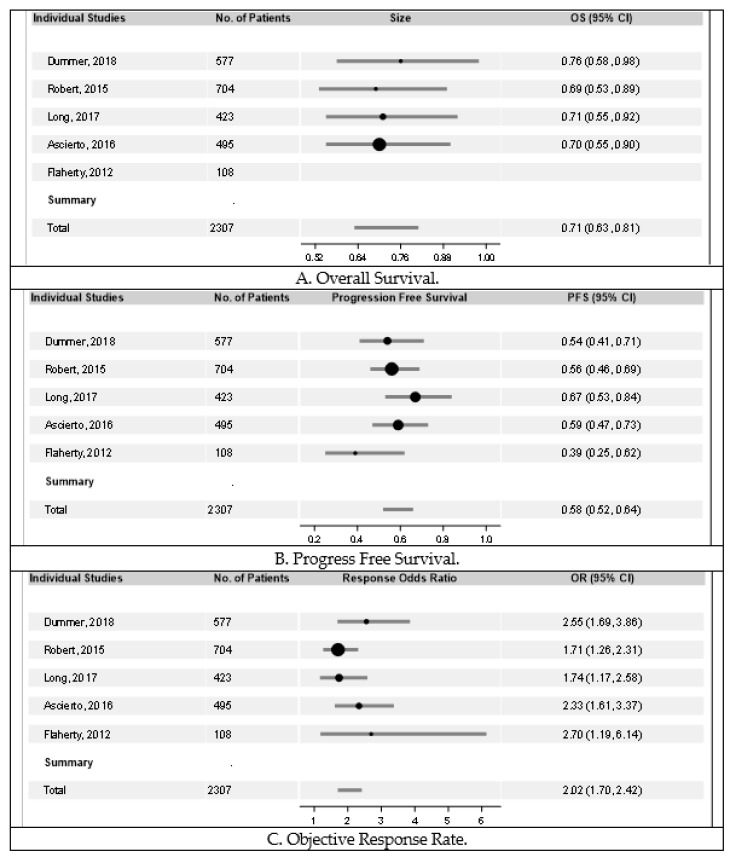
Forrest plots of overall survival, progression-free survival, objective response rate, respectively. Overall survival (OS) and progression-free survival (PFS) were estimated using hazard ratios (HR). Objective response rate (ORR) was measured using odds ratio (OR). The bars in the figures indicate the respective confidence intervals. The sizes of the circles represent the weight of the respective study in the analysis. As demonstrated in (**A**), significant benefit was demonstrated in OS in groups receiving combination therapy in all individual studies and with a combined HR of 0.71 (95% CI 0.63–0.81). Cochran’s Q for OS was 0.31 (*p* = 0.96). Represented in (**B**), significant benefit was demonstrated in PFS in groups receiving combination therapy in all individual studies and with a combined HR of 0.58 (95% CI 0.52–0.64). Cochran’s Q for OS was 4.82 (*p* = 0.31). Similarly, in (**C**), significant benefit was demonstrated in ORR in groups receiving combination therapy in all individual studies and with a combined OR of 2.02 (95% CI 1.70–2.42).

**Figure 4 cancers-11-01950-f004:**
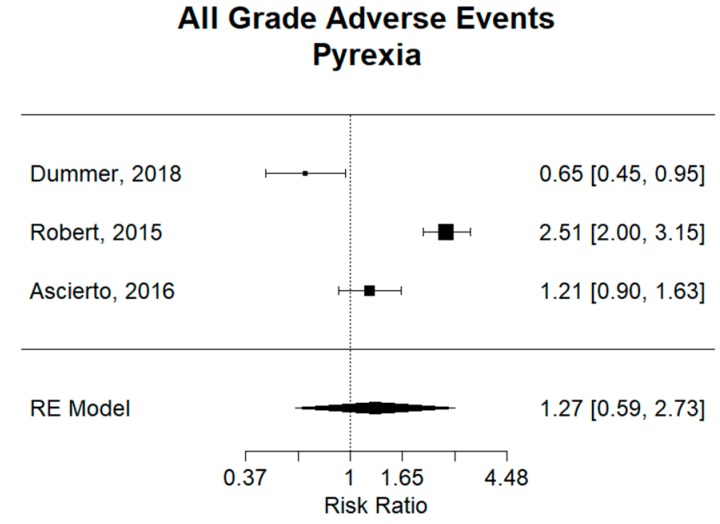
Relative risk (RR) of “pyrexia” adverse events; all grades. Combined RR of all studies was 1.27 (95% CI 0.59–2.73), showing no significant difference in risk of pyrexia between combination and monotherapy. Encorafenib plus binimetinib in the Dummer study had an RR of 0.65 (95% CI 0.45–0.95), revealing a significantly smaller risk of pyrexia as compare to monotherapy. Dabrafenib plus trametinib in the Robert study had an RR of 2.51 (95% CI 2.00–3.15), revealing a significantly higher risk of pyrexia in the combination group. Vemurafenib and cobimetinib compared to vemurafenib in the Ascierto study did not show and significant difference in risk of pyrexia between groups.

**Figure 5 cancers-11-01950-f005:**
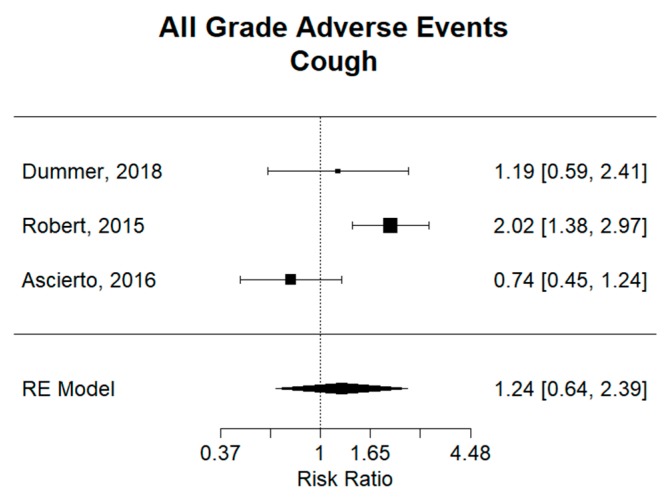
Relative risk (RR) of “cough” adverse events; all grades. Combined RR for all studies was 1.24 (95% CI 0.64–2.39), showing no significant difference in risk of cough between combination and monotherapy. Dabrafenib plus trametinib in the Robert study had an RR of 2.02 (95% CI 1.38–2.97), revealing a significantly larger risk of cough as compared to vemurafenib monotherapy. Encorafenib plus binimetinib in the Dummer study and Vemurafenib plus cobimetinib in the Ascierto study did not show any significant difference in risk of cough compared to vemurafenib monotherapy.

**Figure 6 cancers-11-01950-f006:**
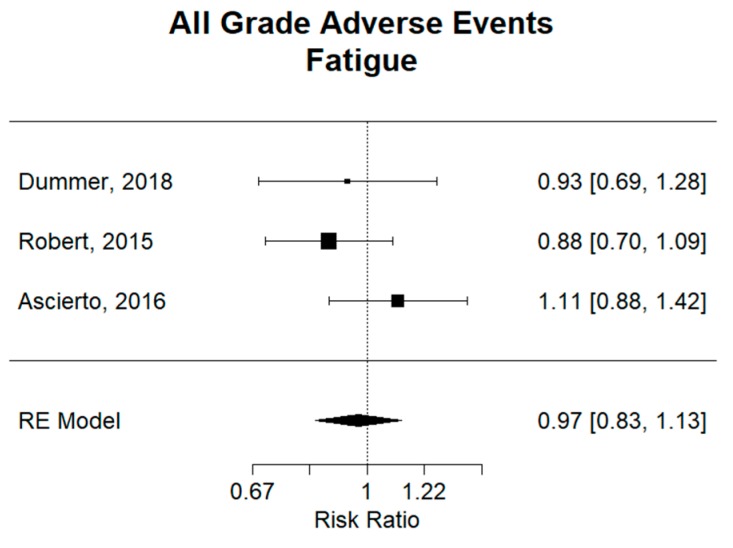
Relative risk (RR) of “fatigue” adverse events; all grades. Combined RR of all studies was 0.97 (95% CI 0.83–1.13), showing no significant difference in risk of fatigue between combination and monotherapy. Relative risk in individual studies also showed no significant difference in risk of fatigue between groups.

**Figure 7 cancers-11-01950-f007:**
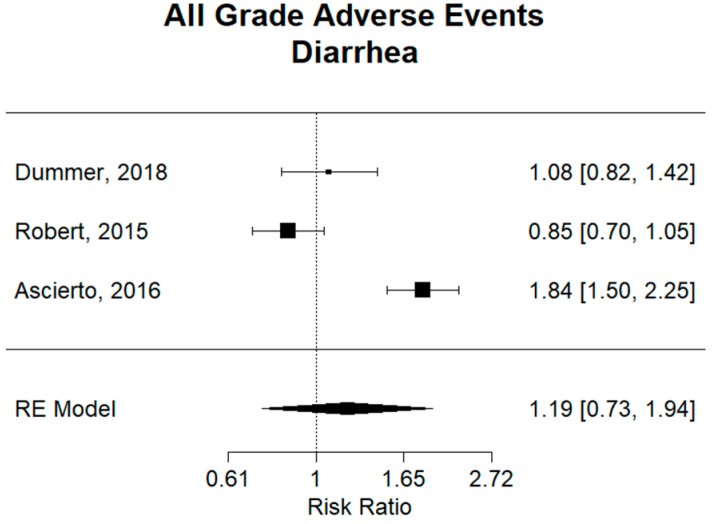
Relative risk (RR) of “diarrhea” adverse events; all grades. Combined RR for all studies was 1.19 (95% CI 0.73–1.94), showing no significant difference in risk of diarrhea between combination and monotherapy groups. There is no significant difference in encorafenib plus binimetinib in the trial by Dummer (RR: 1.08, 95% CI 0.82–1.42) or dabrafenib plus trametinib in the trial by Robert (RR: 0.85, 95% CI 0.70–1.05) when compared to vemurafenib monotherapy. Vemurafenib plus cobimetinib in the trial by Ascierto had a relative risk of 1.84 (95% CI 1.50–2.25), showing a statistically significant high risk of diarrhea when compared to vemurafenib monotherapy.

**Figure 8 cancers-11-01950-f008:**
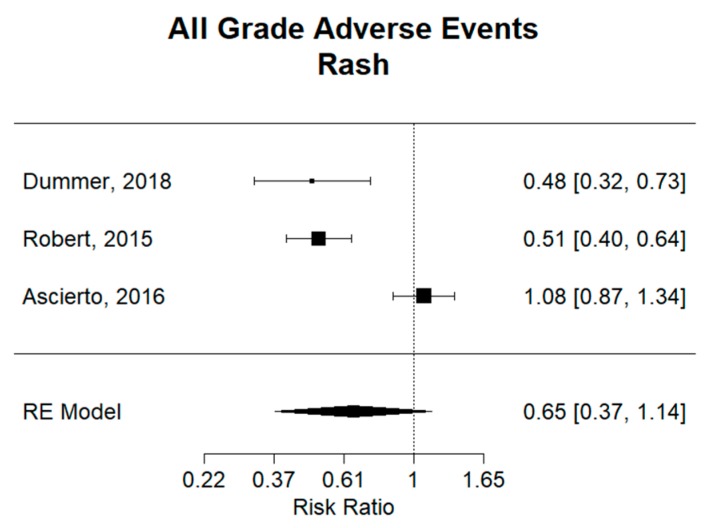
Relative risk (RR) of “rash” adverse events; all grades. Combined RR for all studies was 0.65 (95% CI 0.37–1.14), showing no significant difference in the risk of rash between combination therapy and monotherapy. Combination therapies of encorafenib plus binimetinib in the trial by Dummer (RR: 0.48, 95% CI 0.32–0.73) and dabrafenib plus trametinib in the trial by Robert (RR: 0.51, 95% CI 0.40–0.64) showed significantly decreased risk of rash compared to vemurafenib. Vemurafenib plus cobimetinib in the trial by (RR: 1.08, 95% CI 0.87–1.34) showed no statistical significance compared to vemurafenib.

**Table 1 cancers-11-01950-t001:** Trial characteristics and reported outcomes.

Author	Study Dates	Region	Trial Design	Treatment Regimen	OS (median mos), Combination vs. Monotherapy	PFS (median mos), Combination vs. Monotherapy	ORR, Combination vs. Monotherapy	Follow-up (median mos), Combination vs. Monotherapy
Dummer et al. 2018 [13]	December 2013–April 2015	Worldwide (162 centers, 28 countries)	Phase III encorafenib + binimetinib (*N* = 192) vs. vemurafenib (*N* = 191)	450 mg qd + 45 mg bid vs. 960 mg bid	33.6 vs. 16.9	14.9 vs. 7.3	63% vs. 40%	16.7 vs. 14.4
Robert et al. 2015 [12]	June 2012–October 2013	Worldwide (193 centers)	Phase III dabrafenib + trametinib (*N* = 352) vs. vemurafenib (*N* = 352)	150 mg bid + 2 mg qd vs. 960 mg bid	(NA), 17.2	11.4 vs. 7.3	64% vs. 51%	11 vs. 19
Long et al. 2017 [11]	May 2012–November 2012	Worldwide (113 centers, 14 countries)	Phase III dabrafenib + trametinib (*N* = 211) vs. dabrafenib (*N* = 212)	150 mg bid + 2 mg qd vs. 150 mg bid	25.1 vs. 18.7	11.0 vs. 8.8	68% vs. 55%	9 (all groups)
Ascierto et al. 2016 [9]	January 2013–January 2014	Worldwide (135 centers, 19 countries)	Phase III vemurafenib + cobimetinib (*N* = 247) vs. vemurafenib (*N* = 248)	960 mg bid + 60 mg qd vs. 960 mg bid	22.3 vs. 17.4	12.3 vs. 7.2	70% vs. 50%	14.2 (all groups)
Flaherty et al. 2012 [10]	March 2010–July2011	Multi-national (16 centers)	Phase I/II dabrafenib + trametinib (*N* = 54) vs. dabrafenib (*N* = 54)	150 mg bid + 2 mg qd vs. 150 mg bid	(NA)	9.4 vs. 5.8	76% vs. 54%	14.1 (all groups)

NA = not available; OS = overall survival; PFS = progression-free survival; ORR = objective response rate; mos = months.

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
