# Peer review of "Efficacy and Adverse Events in Metastatic Melanoma Patients Treated with Combination BRAF Plus MEK Inhibitors Versus BRAF Inhibitors: A Systematic Review"

_cancers, 2019, doi:10.3390/cancers11121950_

Round 1

Reviewer 1 Report

I have read a review manuscript (ID: cancers-592492) entitled “Efficacy, Adverse Events in Melanoma Patients Treated with Combined BRAF plus MEK inhibitors versus BRAF Inhibitors: A Systematic Review and Meta-analysis” written by Austin Greco et al.

This articles is a systemic review with metaanalysis based on literature search strategy to select published papers available in PubMed, Cochrane Database of Systematic Reviews, ASCO and EMBASE data bases.

This analysis has some merits which makes the manuscript interesting for some specific reader, but the originality and scientific impact of this study is not difficult to accept. The main bias of this study metaanlysis is that the authors included to the analysis two studies published in two papers (Long et al. Ann Oncol 2017, and Robert et al. NEJM 2015) which are in fact the arms of the same randomized study registered with ClinicalTrials:  BRF113220, number NCT01072175; COMBI-d, number NCT01584648; COMBI-v, number NCT01597908.

Taking consideration that all data obtained within these clinical trials are available on https://clinicaltrials.gov/, and no “data mining” approach was necessary to result the obtained databases. All information about outcomes are published on this webpage.

Major comments:

In the meta analysis both a phase I/II and a phase III clinical trials were included. Wherein only one clinical trial among 5 (20%) (Flaherty NEJM 2012 ), which included overall 108 patients (5%) has been included. Because of different patients inclusion criteria among the phase I and other phases, the analysis may result spurious data.

The follow-up period has neither been mentioned nor analyzed.

It is an approximation study, the authors lack the opportunity to evaluate a hypothesis e.g. that combined therapy with MRK and BRAF inhibitors can have various effects depending on : cohort selection, demographic (age, gender) and geographic (origin) factors (see: DerSimonian Contemp Clin Trials. 2015).

Poor quality of graphics, currently number of graphical software is used to visualize data in more eye-catching manner. Drawings and plots should be designed in panels of presenting different adverse events in comparison.

Conclusions are trivial.

English style needs major correction.

Author Response

We agree with your note to categorize this as a systematic review rather than meta-analysis without the hypothesis. Thank you for this note.

In terms of the graphics, I hope the improved descriptions of the figures helps. The statistician that made this has left the institution. We will discuss with the reviewers to determine what visual representation would be better. 

The english style has been corrected. 

Thank you very much for the review. 

Reviewer 2 Report

Comments :

The metanalysis provides an overview of BRAF and MEK inhibitors safety and toxicity profile over BRAF inhibitors alone.

However some questions need to be addressed:

- why decrease LVEF and ophthalmologic toxicity which are common limiting toxicities of MEK inhibitor, and require close cardiac and ophthalmologic examination during treatment, are not studied?

- vemurafenib is the only comparator used to compare toxicities of combined BRAF and MEK inhibitors in the toxicity section but it is considered as the most toxic drug among BRAF inhibitors. This method limits the extrapolation of the data.

Similarly I don’t understand why for the study by Dummer [14], the participants in the mono-therapy group treated with encorafenib were excluded as usually in every study, BRAF and MEK inhibitors are compared and associated by pairs (vemu with cobi, enco with bini etc). We are loosing some interesting data.

- pyrexia is not the only toxicity that should be considered to choose between enco bini and dabra trame. The conclusion is too extrapolated and does not reflect the whole results of the study “Most notably, encorafenib plus binimetinib group had a significantly smaller risk of pyrexia than the combination of dabrafenib plus trametinib. Changing combination therapy to encorafenib plus binimetinib for patients experiencing pyrexia on dabrafenib plus trametinib could be an appropriate option.”

It should be rephrased

- intro: « far less favorable at 23% for metastatic disease” : without treatment ? because with the treatments now used in melanoma this rate is not correct anymore.

- “side effect profiles are proposed to be improved with combination therapy, which may be secondary to development of additional mutations in the MAPK pathway while on mono-therapy ”: it is the efficacy that is improved by the addition of MEK inh to BRAF inh to prevent the development of secondary mutation

- there are more than 40% of mutation in the BRAF gene in melanoma but 40% of BRAF V600 activating mutations. This should be corrected throughout the manuscript.

Author Response

The metanalysis provides an overview of BRAF and MEK inhibitors safety and toxicity profile over BRAF inhibitors alone.

However some questions need to be addressed:

- why decrease LVEF and ophthalmologic toxicity which are common limiting toxicities of MEK inhibitor, and require close cardiac and ophthalmologic examination during treatment, are not studied?

Unfortunately, it was decided that ophthalmologic toxicities would not be evaluated. Also, certain cardiac toxicities such as heart failure was not included as the incidence may not have been reported in 1 of the 3 studies comparing to vemurafenib monotherapy and the analysis was not power for only comparing 2 studies in the tables. The latter was described in section 2.3.

- vemurafenib is the only comparator used to compare toxicities of combined BRAF and MEK inhibitors in the toxicity section but it is considered as the most toxic drug among BRAF inhibitors. This method limits the extrapolation of the data.

Similarly I don’t understand why for the study by Dummer [14], the participants in the mono-therapy group treated with encorafenib were excluded as usually in every study, BRAF and MEK inhibitors are compared and associated by pairs (vemu with cobi, enco with bini etc). We are loosing some interesting data.

We discussed this option thoroughly to include the arm comparing to encorafenib monotherapy, however, we felt that having data only comparing to vemurafenib monotherapy allowed to better direct comparison between combination groups.  This is discussed in section 2.5 as well.

- pyrexia is not the only toxicity that should be considered to choose between enco bini and dabra trame. The conclusion is too extrapolated and does not reflect the whole results of the study “Most notably, encorafenib plus binimetinib group had a significantly smaller risk of pyrexia than the combination of dabrafenib plus trametinib. Changing combination therapy to encorafenib plus binimetinib for patients experiencing pyrexia on dabrafenib plus trametinib could be an appropriate option.”

It should be rephrased

I just included this as notably, rather than most notably. I have similarly worded conclusive statements in the discussion that I believe would make the conclusion too long to repeat them all. Please let me know if you disagree and I can make this change. 

- intro: « far less favorable at 23% for metastatic disease” : without treatment ? because with the treatments now used in melanoma this rate is not correct anymore.

Updated to state: “with previous treatment methods”

- “side effect profiles are proposed to be improved with combination therapy, which may be secondary to development of additional mutations in the MAPK pathway while on mono-therapy ”: it is the efficacy that is improved by the addition of MEK inh to BRAF inh to prevent the development of secondary mutation

I took this from a basic science paper (source 13). They proposed that paradoxical amplification of MAPK pathway can vary therapies and can also lead to differing toxicities, not just efficacy. I reworded this to hopefully make it more clear.

- there are more than 40% of mutation in the BRAF gene in melanoma but 40% of BRAF V600 activating mutations. This should be corrected throughout the manuscript.

This was corrected

Reviewer 3 Report

This is a well outlined meta-analysis of phase I, II, and III clinical trials comparing treatment efficacy and side effects of BRAF vs BRAF/MEK inhibition in melanoma. The meta-analysis is well written. The methods section was complete and result section comprehensive. Overall, the conclusions are very well supported by the data and figures. 

I only have very minor suggestions for improvement:

Figure 2: Change "Eligability" to "Eligibility".

In section 3.2: Change "When compare" to "When compared" twice in the section. 

Author Response

The errors you noted have been corrected. I appreciate the review.